# Vindoline Exhibits Anti-Diabetic Potential in Insulin-Resistant 3T3-L1 Adipocytes and L6 Skeletal Myoblasts

**DOI:** 10.3390/nu15132865

**Published:** 2023-06-24

**Authors:** Beegum Noorjahan Shijina, Achuthan Radhika, Sainulabdeen Sherin, Prabath Gopalakrishnan Biju

**Affiliations:** 1Department of Biochemistry, University of Kerala, Kariavattom Campus, Thiruvananthapuram 695581, Kerala, India; shijinabeegum01@gmail.com; 2Department of Biochemistry, Government College, Kariavattom, Thiruvananthapuram 695581, Kerala, India; radhikanair02@gmail.com; 3Department of Biochemistry, PMS College of Dental Science and Research, Vattappara, Thiruvananthapuram 695028, Kerala, India; honeyhill86@gmail.com

**Keywords:** obesity, T2DM, insulin resistance, vindoline, adipocyte, skeletal muscle

## Abstract

Type 2 diabetes mellitus (T2DM) emerged as a major health care concern in modern society, primarily due to lifestyle changes and dietary habits. Obesity-induced insulin resistance is considered as the major pathogenic factor in T2DM. In this study, we investigated the effect of vindoline, an indole alkaloid of *Catharanthus roseus* on insulin resistance (IR), oxidative stress and inflammatory responses in dexamethasone (IR inducer)-induced dysfunctional 3T3-L1 adipocytes and high-glucose-induced insulin-resistant L6-myoblast cells. Results showed that dexamethasone-induced dysfunctional 3T3-L1 adipocytes treated with different concentrations of vindoline significantly enhanced basal glucose consumption, accompanied by increased expression of GLUT-4, IRS-1 and adiponectin. Similarly, vindoline-treated insulin-resistant L6 myoblasts exhibited significantly enhanced glycogen content accompanied with upregulation of IRS-1 and GLUT-4. Thus, in vitro studies of vindoline in insulin resistant skeleton muscle and dysfunctional adipocytes confirmed that vindoline treatment significantly mitigated insulin resistance in myotubes and improved functional status of adipocytes. These results demonstrated that vindoline has the potential to be used as a therapeutic agent to ameliorate obesity-induced T2DM-associated insulin resistance profile in adipocytes and skeletal muscles.

## 1. Introduction

Insulin resistance (IR), an impaired response of peripheral tissues to insulin, is characterized by hyperinsulinemia, hyperglycemia, hypertension, dyslipidemia, visceral adiposity, hyperuricemia, elevated inflammatory markers, endothelial dysfunction and a prothrombic state. Chronic insulin resistance can result in metabolic syndrome, non-alcoholic fatty liver disease (NAFLD) and type 2 diabetes mellitus (T2DM). T2DM is the predominant consequence of insulin resistance. Muscle, liver and adipose tissue are the three primary sites of insulin resistance [1].

According to previous studies, one of the major causes of IR is chronic obesity [2]. Obesity associated IR is closely linked to adipocyte dysfunction [3], resulting in impairment of glucose and lipid homeostasis as well as inflammatory responses [4]. Adipose tissue regulates both glucose and lipid metabolism by releasing adipokines, pro-inflammatory cytokines, and free fatty acids. In respond to insulin, adipose tissue enhance differentiation of preadipocytes into adipocytes, storage of triglycerides, inhibiting lipolysis and thereby promoting the uptake of glucose and free fatty acids [5].

Under chronic obese condition, increased free fatty acids induce cellular oxidative stress in adipose tissue through the production of reactive oxygen species (ROS), which trigger adipocyte dysfunction and induce production of pro-inflammatory cytokines such as TNF-α, IL-6 and IL-1β from adipose tissue, consequentially potentiating adipose tissue inflammation and obesity-related diseases such as T2DM [6]. Adipose dysfunction in obesity also potentiates insulin resistance and triglyceride storage impairment. These abnormalities lead to increased accumulation of free fatty acids in circulation and increased accumulation of free fatty acids in skeleton muscle, triggering insulin resistance in skeleton muscle [6]. In humans, approximately 80% of postprandial glucose disposal is through the skeletal muscle and their proper response to insulin action is vital to maintaining glucose homeostasis [7]. In a fed state, the normal blood glucose disposal in the skeletal muscle is maintained by GLUT4 translocation to the plasma membrane and the failure of this event in response to insulin shows an early stage of insulin resistance and T2DM [8]. These findings indicated that therapeutic strategies ameliorating adipocyte dysfunction and related skeleton muscle insulin resistance may be beneficial in the management and treatment of obesity-related T2DM.

*Catharanthus roseus*, an evergreen plant belonging to the apocynacea family, is a chief source of indole alkaloids [9]. Our own previous studies on different solvent fractions from the leaves of *C. roseus* established their non-toxic nature and anti-diabetic properties. Activity driven sequential extraction of the different fractions identified vindoline, an indole alkaloid found in high concentrations in the leaves of *C. roseus,* as the most significant contributor of the observed effects in insulin-resistant (IR) adipocytes. Previous studies confirmed that vindoline exhibited anti-diabetic properties of the plant [10,11]. Moreover, a previous study in T2DM rats demonstrated that vindoline possess anti-hyperglycemic, anti-hyperlipidemic, anti-inflammatory and antioxidant properties [9]. The model used in this study was a low-dose streptozotocin-induced Wistar rat model. However, the effect of vindoline on IR models such as dysfunctional adipocytes and skeletal muscles were not examined so far. Hence, in this study, we evaluated the anti-diabetic properties of vindoline in dexamethasone-induced dysfunctional 3T3-L1 adipocytes and in high-glucose-induced IR L6 skeleton myoblasts.

## 2. Materials and Methods

### 2.1. Chemicals, Biochemicals and Cell Lines

All chemicals and reagents used were of analytical grade. 3-(4, 5-dimethylthiazol-2-yl)-2, 5-diphenyltetrazolium bromide (MTT), oil red O stain, dichloro-dihydro-fluorescein diacetate (DCF-DA), 3-isobutyl-1-methyl xanthine (IBMX), insulin, TRIzol and dexamethasone were purchased from Sigma Aldrich, (St. Louis, MO, USA). Dulbecco’s Modified Eagle’s Medium (DMEM), fetal bovine serum (FBS) and antibiotic solutions were procured from Gibco, Thermo Fisher Scientific, (Waltham, MA, USA). 3T3-L1 pre-adipocyte and L6-myoblast cell lines were obtained from National Centre for Cell Science (NCCS), Pune, India. All other chemicals and solvents were purchased from SRL (Maharastra, India), Ranbaxy (Delhi, India), Merck (Darmstadt, Germany) and Spectrochem (Mumbai, India).

### 2.2. Plant Material and Extraction Procedure

*C. roseus* leaves were collected from Thiruvananthapuram district, Kerala, India during the month of October. The specimen was authenticated by an expert (Dr. Valsaladevi, Curator, Department of Botany, University of Kerala) and a specimen voucher number was obtained (KUBH 10135). The washed and air-dried leaves were made into powder in a mechanical blender and subjected to hot methanolic extraction with a Soxhlet apparatus. Extract was evaporated to dryness using a rotary flash evaporator.

### 2.3. Isolation of Vindoline

For the isolation of vindoline, methanolic extract of *C. roseus* leaves was subjected to silica gel column chromatography using step gradient elution from 100% dichloromethane (DCM) to 100% methanol and eluted fractions were collected. Based on the thin layer chromatography (TLC) results under Dragendroff’s reagent, fraction-3 was subjected to silica sub column chromatography using DCM and methanol and eluted sub-fractions were collected. The presence of vindoline was confirmed by LC-MS/MS analysis (Shimadzu, Kyoto, Japan) [12].

### 2.4. 3T3-L1 Cell Culture, Differentiation and Induction of Dysfunction in Adipocytes

Mouse 3T3-L1 preadipocytes were cultured in DMEM (catalogue no. 11995065) supplemented with heat-inactivated 10% FBS plus antibiotic mixture (100U penicillin/mL and 100 µg streptomycin/mL) in a humidified atmosphere of 5% CO_2_ at 37 °C. Cells at 80% confluence were allowed to differentiate from preadipocytes to fully matured adipocytes following the already established protocol [13].

Briefly, 80% confluence pre-adipocytes were cultured in adipogenic medium containing 0.5 mM 3-isobutyl-1-methyl xanthine (IBMX), 10–20 µg/mL insulin, 1 µM dexamethasone, 10% heat-inactivated FBS, antibiotic mixture in DMEM (catalogue no. 11995065) for 3 days to induce differentiation to adipocytes. After 3 days, this medium was replaced with post differentiation medium containing only 10–20 µg/mL insulin in 10% FBS/DMEM for 2 days. Fully differentiated cells were used for the subsequent studies

For the induction of adipocyte dysfunctions, 12 h serum starved mature adipocytes were incubated for 48 h with 1 µM dexamethasone and 1 µM insulin [4] and these dysfunctional 3T3-L1 adipocytes were grouped as follows.

Group 1: normal mature adipocytes (3T3-control); Group 2: untreated dysfunctional adipocytes (model); Group 3: dysfunctional adipocytes + 12.5 µg/mL vindoline (VDL 12.5); Group 4: dysfunctional adipocytes + 25 µg/mL vindoline (VDL 25); Group 5: dysfunctional adipocytes + 1 mM metformin (Met). The treatment period was 24 h.

### 2.5. L6-Myoblast Cell Culture and Induction of Insulin Resistance

Rat skeletal muscle L6 myoblast cells were differentiated to L6 myotubes by the method of Gao et al., 2009 [14]. Briefly, myoblast cells maintained in high glucose DMEM medium [DMEM, High glucose: 4.5 g/L D-glucose, L-glutamine, 110 g/L sodium pyruvate (catalogue no. 11995065) supplemented with heat-inactivated 10% FBS plus antibiotic mixture (100U penicillin/mL and 100 µg streptomycin/mL) were passaged at 80% confluence to differentiation medium. Differentiation was carried out in low glucose DMEM medium [DMEM, low glucose: 1 g/L D-glucose, L-glutamine, sodium bicarbonate and 110 g/L sodium pyruvate. Catalogue no. 12320032)] supplemented with 2% FBS for three days, and the differentiation from L6 myoblasts to myotubes was confirmed by the formation of multinucleation in the cells.

After differentiation, myotubes were incubated in DMEM containing 25 mM/L glucose for 4–5 days to induce insulin resistance [15], and these insulin-resistant myotubes (IR-myotubes) were grouped as follows.

Group 1: normal myotubes (Control); Group 2: untreated IR-myotubes (Model); Group 3: IR-myotubes + 12.5 µg/mL vindoline (VDL 12.5); Group 4: IR-myotubes + 25 µg/mL vindoline (VDL 25); Group 5: IR-myotubes + 1 mM metformin (Met). The treatment period was 24 h.

### 2.6. Evaluation of Cell Viability

Cell viability of 3T3-L1 adipocytes and L6 myoblasts when treated with different concentrations (12.5, 25, 50, 75 and 100 µg/mL) was carried out by MTT assay protocol [16] using a commercial assay kit (Thermo Fisher Scientific, USA) in 96 well plates. Cells were incubated for 24 h at under saturated humidity at 37 °C and 5% CO_2_. After incubation period, media were aspirated, followed by addition of fresh media along with 20 µL MTT (5 mg/mL) solution/well. The yellowish MTT was reduced to dark-colored formazan by viable cells only. The formazan crystals formed were solubilized with MTT lysis buffer. The color developed was quantified with an ELISA plate reader (Bio-Rad, Herculus, CA, USA) (measuring wavelength: 570 nm). A graph was plotted by taking percentage viability in the Y-axis and concentration of vindoline in the X-axis.

### 2.7. Evaluation of Glucose Consumption Activity

The glucose consumption ability of dysfunctional adipocytes and IR-myotubes after vindoline treatment was analyzed according to a previously described method [17]. Briefly, the dexamethasone plus insulin-induced dysfunctional adipocytes or IR-myotubes were preincubated with DMEM containing 0.2% FBS for 12 h and were then incubated with different concentrations (12.5 and 25 µg/mL) of vindoline for 24 h. The medium was collected and glucose concentration were determined by the glucose oxidase method using a commercial kit (Agappe Diagnostics, Kerala, India). Briefly, 10 µL sample or glucose standard was added to 1 mL glucose reagent containing 92 mmol/L tris buffer (pH 7.4), 0.3 mmol/L phenol, 15,000 U/L glucose oxidase and 2.6 mmol/L 4-aminophenazone and incubated for 10 min at 37 °C. The absorbance was measured at 505 nm using a microplate reader. The amount of glucose consumption was calculated as the difference between glucose concentrations of blank wells and wells with cells under treatment.

### 2.8. Evaluation of Intracellular Lipid Accumulation

The fat accumulation in dexamethasone-induced dysfunctional adipocyte with and without vindoline treatment was determined using Oil red O assay. Briefly, the dexamethasone plus insulin-induced dysfunctional adipocytes plated in 24 well plates were treated with different concentrations of vindoline for 24 h at 37 °C. Cells were washed with PBS twice and fixed with 10% formalin. After 30 min fixation, formalin was removed and cells were washed twice with distilled water. To this, freshly prepared oil red O solution (0.5% oil red O in isopropanol) was added and kept for 60 min at room temperature. After incubation, cells were washed with PBS to remove extra dye for the capturing the image. Isopropanol was added to extract the stain from cells and absorbance measured at 500 nm using a microplate reader.

### 2.9. Evaluation of Glycogen Content

The glycogen content of IR-myotubes after vindoline treatment was estimated according to previous reports [18]. Briefly, IR myotubes were treated with different concentrations of vindoline for 24 h. Then, 1 mM metformin and 10 μL of 10 nM insulin were used as positive control and stimulant, respectively. After removal of medium, HEPES buffer was added and kept for 2 min, followed by addition of 10 mM glucose in HEPES buffer and incubated for 30 min. After incubation, media was aspirated followed by addition of 30% potassium hydroxide and kept for 5 min to lyse the cells. The lysate was heated for 30 min at 70 °C and chilled on ice along with addition of ice cold 95% ethanol and left overnight at 80 °C for glycogen precipitation. The content was centrifuged at 8000× *g* for 10 min and pellet collected. To this, anthrone reagent was added and cooled in ice for 10 min. The tubes were heated for 10 min at 80 °C and cooled. The absorbance was measured at 620 nm using a microplate reader [19].

### 2.10. gGene Expression Studies

For gene expression studies, total RNA was isolated from mouse 3T3-L1 adipocytes and rat L6-myotubes using TRIzol reagent according to the manufacturer’s instructions and cDNA synthesis was performed using Bio-Rad cDNA synthesis kit (I Script reverse transcription super mix for RT-PCR). The real time qPCR was carried out using Bio-Rad RT-PCR kit (iTaqTM Universal SYBR green super mix) on real time PCR instrument according to manufacturer’s protocol.

The following gene specific primers were used (Table 1):

### 2.11. Statistical Analysis

Values are expressed as mean ± standard deviation (SD) of triplicate values of three independent experiments for each sample within a parameter. Graphical representation and statistical analysis were carried out on GraphPad Prism 9.5 software (San Diego, CA, USA). Statistical analysis was carried out by one-way ANOVA with Tukey’s post hoc multiple comparison test. Statistically significant difference for *p* < 0.05 is indicated by different symbols, as mentioned in figure legends.

## 3. Results

### 3.1. Isolation of Vindoline

LC/MS-MS analysis of sub-fraction 3 obtained from sub column chromatography showed base peak at 457, indicative of the presence of the indole alkaloid vindoline (Figure 1a). MS/MS fragmentation pattern of the sub-fraction 3 showed ions at *m*/*z* 187.9, 411, 439, 457, etc., related to structure of vindoline (Figure 2b), confirming that the isolated compound was purified vindoline.

### 3.2. Effect of Vindoline on 3T3-L1 Adipocyte and L6 Myotube Viability

The MTT assay results showed that vindoline treatment up to 25 µg/mL for 24 h did not exhibit any significant effect on viability of 3T3-L1 adipocytes and L6 myotubes with respect to untreated cells, respectively (Figure 2). Hence, we used 12.5 and 25 µg/mL vindoline for subsequent studies.

### 3.3. Effect of Vindoline on Adipocyte Dysfunction

#### 3.3.1. Effect of Vindoline on Glucose Consumption in Dysfunctional Adipocytes

The glucose consumption of dexamethasone-induced dysfunctional adipocytes was decreased significantly with respect to healthy adipocytes. Treatment with vindoline for 24 h significantly increased the glucose consumption of dysfunctional adipocytes in a dose-dependent manner (Figure 3).

#### 3.3.2. Effect of Vindoline on Lipid Accumulation in Dysfunctional Adipocytes

Intracellular lipid accumulation, which is characteristic of mature adipocytes, was found to decrease after the induction of dysfunction with dexamethasone for 48 h. However, a significant increase in lipid content was observed in dysfunctional adipocytes after vindoline treatment in a dose-dependent manner, which was also confirmed by quantification method (Figure 4a,b).

#### 3.3.3. Effect of Vindoline on GLUT-4, IRS-1 and Adiponectin Expression in Dysfunctional Adipocytes

The mRNA expression levels of GLUT-4, IRS-1 and adiponectin were significantly lower in dexamethasone-induced dysfunctional adipocytes compared with healthy adipocytes. Dysfunctional adipocytes treated with vindoline showed significantly increased expressions of GLUT-4, IRS-1 and adiponectin (Figure 5).

### 3.4. Effect of Vindoline on IR-L6 Myotubes

#### 3.4.1. Effect of Vindoline on Glucose Consumption in IR-Myotubes

Effect of vindoline on glucose consumption in IR-myotubes was estimated by glucose-oxidase method. As shown in Figure 6, the glucose consumption was significantly decreased in IR model group when compared to control. However, treatment with vindoline for 24 h significantly increased the glucose consumption of model group in a dose-dependent manner.

#### 3.4.2. Effect of Vindoline on Glycogen Content in IR Myotubes

The effect of vindoline on glycogen content was estimated by anthrone method. Compared with the control myotubes, the glycogen content in myotubes induced with high glucose decreased significantly. Treatment with different doses of vindoline in high-glucose-induced myotubes showed a significant increment in glycogen content in a dose-dependent manner and which was comparable to the metformin (Figure 7).

#### 3.4.3. Effect of Vindoline on GLUT-4 and IRS-1 Expression in IR-Myotubes

The mRNA expression levels of GLUT-4 and IRS-1 were significantly lower in glucose-induced IR-myotubes when compared with control. On the other hand, IR-myotubes treated with vindoline showed significantly increased expressions of GLUT-4 and IRS-1 (Figure 8).

## 4. Discussion

Nutritional overload associated with obesity is associated with the onset of T2DM. One of the first tissues of the body to respond to a nutritional overload is adipose tissue [20] and the resultant mediators influence the insulin sensitivity of other tissues [21]. Hence, the main risk factor for insulin resistance is a dysfunctional adipose tissue [22]. While investigations on several agents with antidiabetic properties were carried out, very few studies explored the potential of adipose tissue specific modulatory agents. Vindoline was demonstrated to possess antidiabetic properties [23]. However, no systemic work was carried out on the antioxidant, anti-inflammatory and anti-diabetic effect of vindoline in IR models involving adipocytes and skeletal myoblasts. Hence, this study in which we isolated vindoline from *Catharanthus roseus* leaves confirmed its purity by LC-MS/MS analysis, and carried out antidiabetic property evaluation in IR models assumes significance.

In the first phase of this study, we investigated the role of vindoline on dysfunctional adipocytes using dysfunctional mouse 3T3-L1 adipocytes as an in vitro model system. An initial MTT assay evaluation was carried out with different concentrations of vindoline on 3T3-L1 preadipocytes for its effect on cell viability. Assay results confirmed that more than 94% of the cells were viable up to 25 µg/mL concentration after 24 h treatment as compared to untreated control. Hence, two concentrations of vindoline, namely 12.5 and 25 µg/mL, were selected for the subsequent in vitro studies.

Adipocyte dysfunction inducer dexamethasone was used to induce IR in 3T3-L1 adipocytes. Dexamethasone treatments prevent translocation of GLUT-4 to the cell surface via altering insulin signaling pathway by inhibiting phosphatidylinositol-3-kinase (PI3K) and serine/threonine protein kinase and induces insulin resistance. Dexamethasone treatment also enhances the activity of hormone sensitivity lipase, leading to increased lipolysis in adipocytes [24]. Obesity linked elevated basal fat cell lipolysis is closely related with insulin resistance. Therefore, the inhibition of adipocyte lipolysis is considered as a promising therapeutic goal for treating insulin resistance and obesity-associated T2DM [25]. In the current study, dexamethasone-induced dysfunctional adipocytes showed a significant increase in lipolysis as compared with normal adipocytes, but treatment with different concentrations of vindoline inhibited dexamethasone-induced lipolysis in a dose-dependent manner in dysfunctional adipocytes.

According to previous studies, the capacity of adipocytes to respond to insulin depends on expression of IRS-1 and GLUT-4 [26]. Insulin-stimulated adipocytes exhibited increased glucose uptake due to the translocation of GLUT-4, a main insulin regulated glucose transporter, from the cytosol compartment to the plasma membrane via the activation of IRS-1-PI3K-AKT pathway. An increased amount of GLUT-4 in adipose tissue is a good marker of systemic insulin sensitivity [21,27] and decreased level of IRS-1 was reported to be involved in insulin resistance [28]. Previous studies reported a significant lowering of IRS1 mRNA in non-insulin-dependent diabetes mellitus (NIDDM) individuals [29] and in adipose tissue of abdominally obese women [26]. Hence, we checked the mRNA expression of IRS-1 and GLUT4 genes to evaluate the antidiabetic efficacy of vindoline. Adipokines, the physiologically active secretory products of adipocytes, have an important role in the metabolism of not only adipose tissue but even the whole body. Hence, these can be used as an indicator for adipose tissue dysfunction and insulin resistance [30]. Adiponectin, an anti-inflammatory adipokine secreted from adipose tissue also promote recruitment of GLUT-4 to the plasma membrane, thereby maximizing insulin’s ability to mediate glucose uptake, while a low level of adiponectin is associated with obesity, insulin resistance and T2DM [31]. Our study results showed that dexamethasone-induced dysfunctional 3T3-L1 adipocytes treated with different concentrations of vindoline significantly enhanced basal glucose consumption, accompanied by increased expression of GLUT-4, IRS-1 and adiponectin.

Previous studies reported that any alteration of glucose transport in adipocytes results in insulin resistance in skeletal muscle [32]. Thus, in the next phase of this study, we investigated the role of vindoline on high-glucose-induced in vitro insulin resistant rat skeletal muscle L6 myotubes cells. L6 skeletal muscle cells were treated with different concentrations of vindoline to assess its effect on cell viability. Similar to the observations with adipocytes, the MTT assay results confirmed that more than 90% of the cells were viable up to 25 µg/mL concentration after 24 h treatment as compared to untreated control. Hence, vindoline concentration of 12.5 and 25 µg/mL was selected for the subsequent in vitro studies with L6 myotubes.

In skeletal muscle of diabetics with insulin resistance, intracellular accumulation of lipid metabolites due to obesity associated adipose tissue dysfunction suppress insulin stimulated IRS-1 tyrosine phosphorylation resulting in inhibition of GLUT-4 translocation to membrane and decreased insulin stimulated skeleton muscle glycogen synthesis [33]. In our study, the glycogen content was diminished in IR-myotubes, but vindoline treatment to IR-myotubes significantly enhanced glycogen content accompanied by increased upregulation of IRS-1 and GLUT-4. While previous studies [9,34] investigated the effect of vindoline on streptozotocin-induced models for T2DM, the model used was not specific for obesity-associated adipocyte inflammatory diabetic complications. Additionally, the effect of vindoline on skeletal muscles are not well investigated. This study brings a new understanding on the effect of vindoline in augmenting the glucose utilization capability and restoration of function of two key affected targets of T2DM, namely, adipose and skeletal muscles.

## 5. Conclusions

This study showed that vindoline could exert an anti-diabetic effect in dexamethasone-induced dysfunctional adipocytes and improve insulin sensitivity in high glucose-induced IR-myotubes, possibly by activation of glycogen storage via IRS-1 and GLUT-4 upregulation. The observed effect of vindoline on two key pathophysiological targets of T2DM was reported for the first time here. Hence, vindoline can be used as a potent therapeutic agent for the management of obesity associated T2DM.

## Figures and Tables

**Figure 1 nutrients-15-02865-f001:**
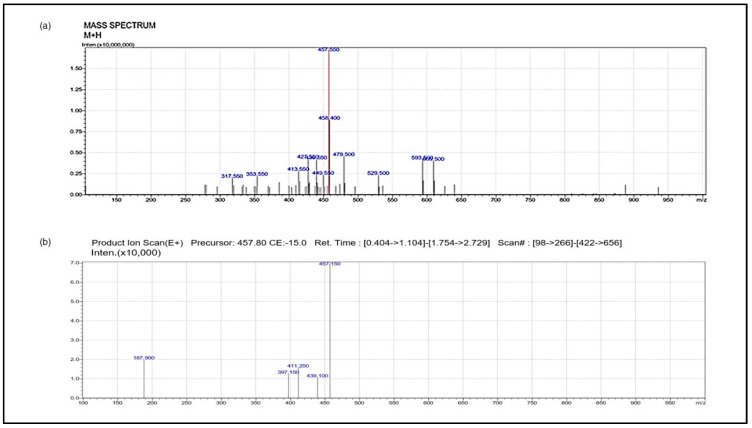
(**a**) Mass spectra of isolated vindoline; (**b**) MS/MS fragmentation pattern of isolated vindoline. LC-MS/MS data for the purified single compound isolated from the leaves of *C. roseus* following successive fractionation and TLC analysis, confirms the presence and purity of vindoline in the isolated compound.

**Figure 2 nutrients-15-02865-f002:**
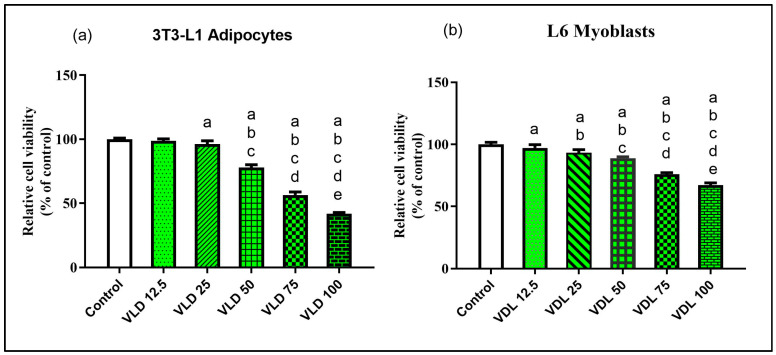
MTT assay. Effect of vindoline on (**a**) 3T3-L1 adipocytes and (**b**) L6 myotubes viability. Cells were treated with vindoline for 24 h. All values are expressed as mean ± SD of triplicate values of three independent experiments for each sample. Statistical comparison at *p* < 0.05. Significant difference is indicated by a different alphabet for comparison between groups. a—significantly different from Control, b—significantly different from VLD 12.5, c—significantly different from VLD 25, d—significantly different from VLD 50, e—significantly different from VLD 75.

**Figure 3 nutrients-15-02865-f003:**
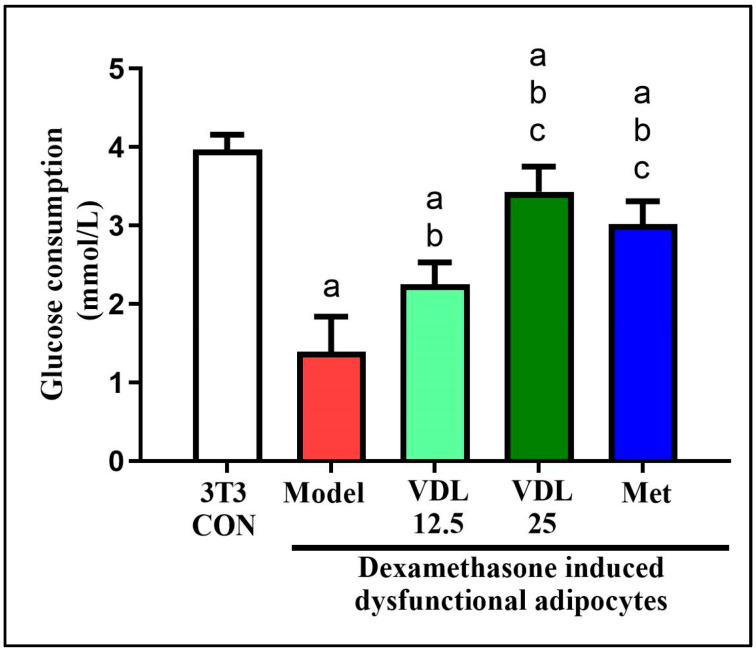
Effect of vindoline on glucose consumption of dexamethasone-induced dysfunctional adipocytes. Grouping are as follows: 3T3-CON indicates normal mature 3T3-L1 adipocytes, Model indicates IR-induced (treated with 1 µM dexamethasone) dysfunctional adipocytes, VDL 12.5 indicates dexamethasone-induced dysfunctional adipocytes treated with 12.5 µg/mL vindoline, VDL 25 indicates dexamethasone-induced dysfunctional adipocytes treated with 25 µg/mL vindoline and Met indicates dexamethasone-induced dysfunctional adipocytes treated with 1 mM metformin. Treatment duration was for 24 h. All values are expressed as mean ± SD of triplicate values of three independent experiments for each sample. Statistical comparison at *p* < 0.05. Significant difference is indicated by a different alphabet for comparison between groups. a—significantly different from 3T3 CON, b—significantly different from Model, c—significantly different from VLD 12.5.

**Figure 4 nutrients-15-02865-f004:**
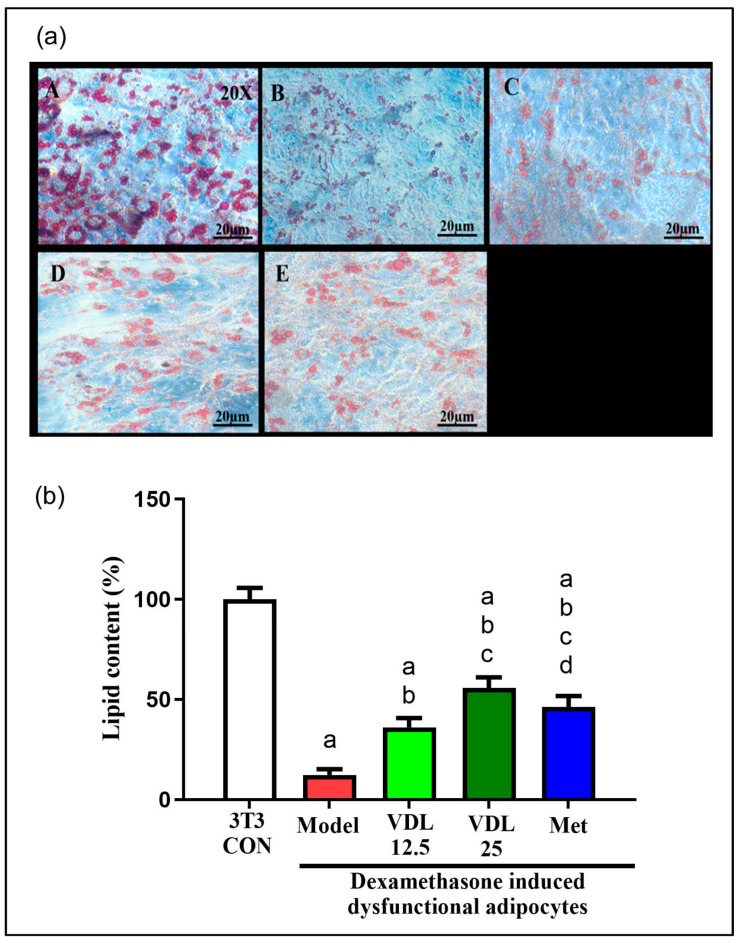
Effect of vindoline on lipid accumulation in dexamethasone-induced dysfunctional adipocytes. (**a**) Images of adipocytes after oil red O staining under 20× magnification. Groups: A—3T3-Con, B—Model, C—VDL 12.5, D—VDL 25, E—Met. (**b**) Quantification of lipid content in dysfunctional adipocytes after vindoline treatment expressed as a percentage of 3T3 control. Grouping are as follows: 3T3-CON indicates normal mature 3T3-L1 adipocytes, Model indicates IR-induced (treated with 1 µM dexamethasone) dysfunctional adipocytes, VDL 12.5 indicates dexamethasone-induced dysfunctional adipocytes treated with 12.5 µg/mL vindoline, VDL 25 indicates dexamethasone-induced dysfunctional adipocytes treated with 25 µg/mL vindoline and Met indicates dexamethasone-induced dysfunctional adipocytes treated with 1 mM metformin. Treatment duration was for 24 h. All values are expressed as mean ± SD of triplicate values of three independent experiments for each sample. Statistical comparison at *p* < 0.05. Significant difference is indicated by a different alphabet for comparison between groups. a—significantly different from 3T3 CON, b—significantly different from Model, c—significantly different from VLD 12.5, d—significantly different from VLD 25.

**Figure 5 nutrients-15-02865-f005:**
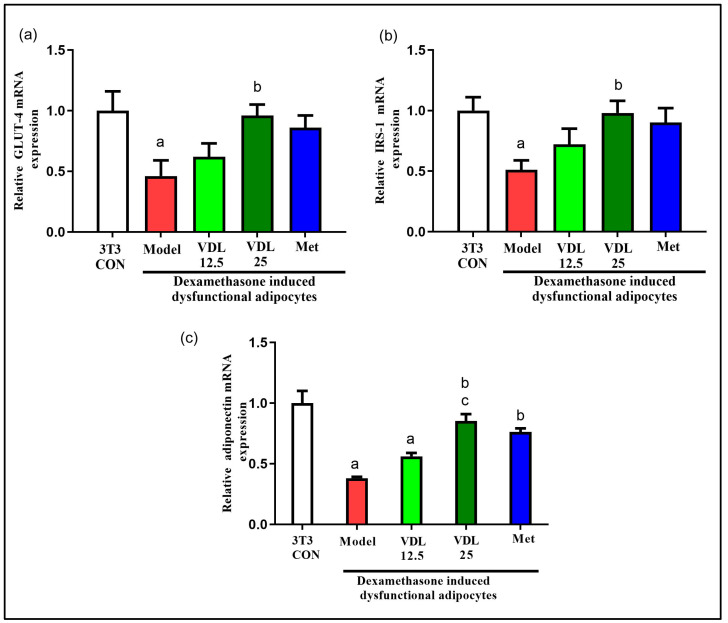
Effect of vindoline on mRNA expression of (**a**) GLUT-4, (**b**) IRS-1 and (**c**) adiponectin in dexamethasone-induced dysfunctional adipocytes. Grouping are as follows: 3T3-CON indicates normal mature 3T3-L1 adipocytes, Model indicates IR-induced (treated with 1 µM dexamethasone) dysfunctional adipocytes, VDL 12.5 indicates dexamethasone-induced dysfunctional adipocytes treated with 12.5 µg/mL vindoline, VDL 25 indicates dexamethasone-induced dysfunctional adipocytes treated with 25 µg/mL vindoline and Met indicates dexamethasone-induced dysfunctional adipocytes treated with 1 mM metformin. Treatment duration was for 24 h. All values are expressed as mean ± SD of triplicate values of three independent experiments for each sample. Statistical comparison at *p* < 0.05. Significant difference is indicated by a different alphabet for comparison between groups. a—significantly different from 3T3 CON, b—significantly different from Model, c—significantly different from VLD 12.5.

**Figure 6 nutrients-15-02865-f006:**
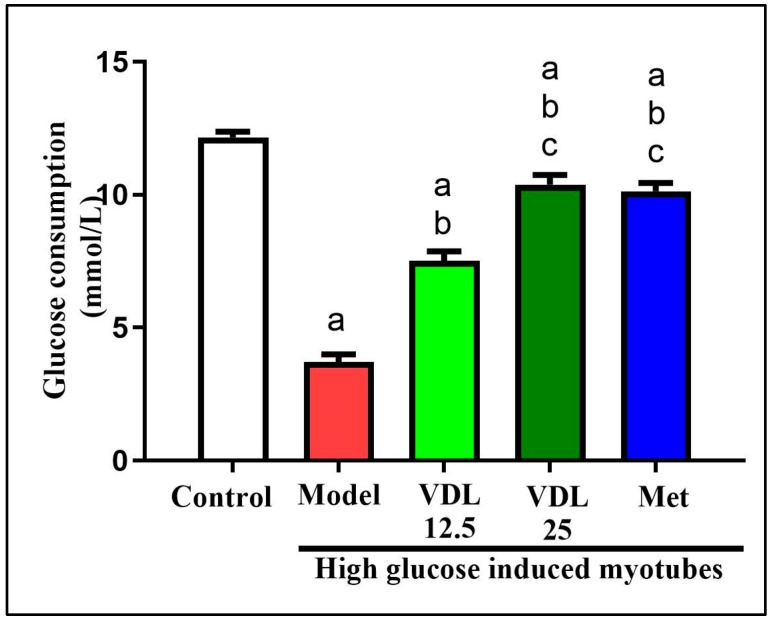
Effect of vindoline on glucose consumption in high glucose-induced IR myotubes. Groupings are as follows: Control indicates normal myotubes, Model indicates untreated high glucose-induced IR myotubes, VDL 12.5 indicates high glucose-induced IR-myotubes treated with 12.5 µg/mL vindoline, VDL 25 indicates high-glucose-induced IR myotubes treated with 25 µg/mL vindoline and Met indicates high glucose-induced IR myotubes treated with 1 mM metformin. Treatment duration was 24 h. All values are expressed as mean ± SD of triplicate values of three independent experiments for each sample. Statistical comparison at *p* < 0.05. Significant difference is indicated by a different alphabet for comparison between groups. a—significantly different from 3T3 CON, b—significantly different from Model, c—significantly different from VLD 12.5.

**Figure 7 nutrients-15-02865-f007:**
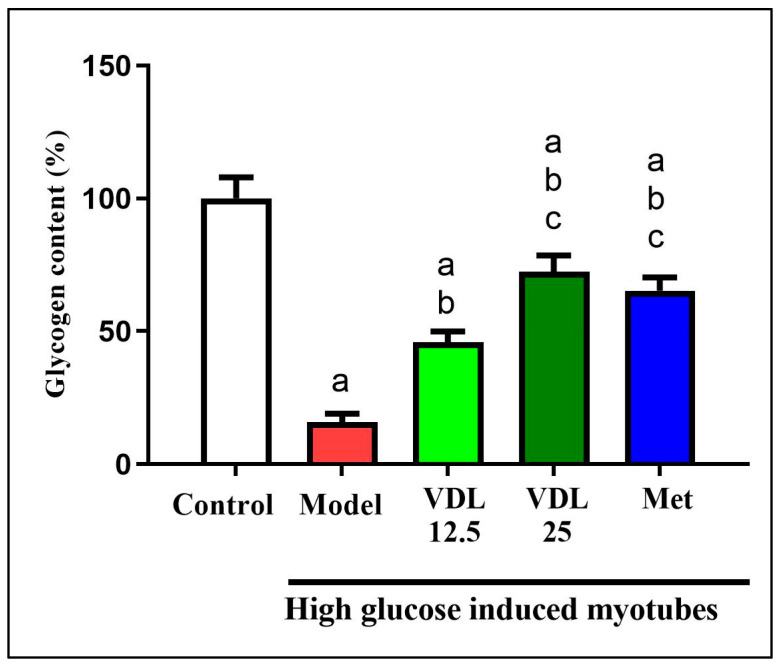
Effect of vindoline on glycogen content in high glucose-induced IR myotubes. Groupings are as follows: Control indicates normal myotubes, Model indicates untreated high glucose-induced IR myotubes, VDL 12.5 indicates high glucose-induced IR myotubes treated with 12.5 µg/mL vindoline, VDL 25 indicates high-glucose-induced IR myotubes treated with 25 µg/mL vindoline and Met indicates high-glucose-induced IR-myotubes treated with 1 mM metformin. Treatment duration was for 24 h. All values are expressed as mean ± SD of triplicate values of three independent experiments for each sample. Statistical comparison at *p* < 0.05. Significant difference is indicated by a different alphabet for comparison between groups. a—significantly different from 3T3 CON, b—significantly different from Model, c—significantly different from VLD 12.5.

**Figure 8 nutrients-15-02865-f008:**
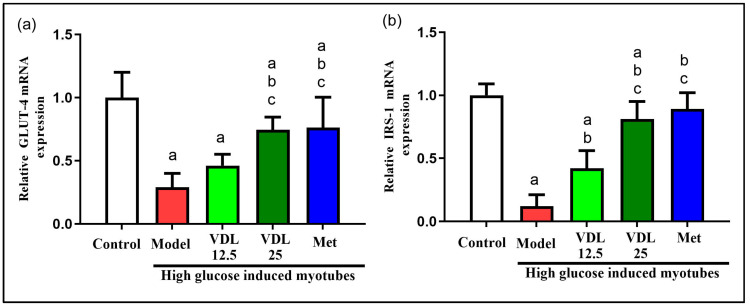
Effect of vindoline on mRNA expression of (**a**) GLUT-4 and (**b**) IRS-1 in high-glucose-induced IR-myotubes. Groupings are as follows: Control indicates normal myotubes, Model indicates untreated high-glucose-induced IR myotubes, VDL 12.5 indicates high-glucose-induced IR myotubes treated with 12.5 µg/mL vindoline, VDL 25 indicates high-glucose-induced IR myotubes treated with 25 µg/mL vindoline and Met indicates high-glucose-induced IR myotubes treated with 1 mM metformin. Treatment duration was for 24 h. All values are expressed as mean ± SD of triplicate values of three independent experiments for each sample. Statistical comparison at *p* < 0.05. Significant difference is indicated by a different alphabet for comparison between groups. a—significantly different from 3T3 CON, b—significantly different from Model, c—significantly different from VLD 12.5.

**Table 1 nutrients-15-02865-t001:** List of primers used for quantitative PCR analysis of transcriptional gene expression of IR-induced 3T3-L1 adipocytes and L6 myotubes treated with vindoline.

Gene	Forward Primer (5′-3′)	Reverse Primer (5′-3′)
Rat IRS-1	CTGCATAATCGGGCAAAGGC	CATCGCTAGGAGAACCGGAC
Rat GLUT-4	GTTGCGGATGCTATGGGTC	GTATGGGGAGTAAGGGAG
Rat β-actin	CCAACCGTGAAAAGATGA	TCCAGTAGTGATAGCCGT
Mice GLUT-4	CTGGCACTTCCACTCAAC	GAGACTGATGCGCTCTAAC
Mice IRS-1	TAACTGGACATCACAGCAGAATG	ACGGATGCATCGTACCATCT
Mice adiponectin	TAAACATTTCCGGCCCCTCC	GCTCCACTGTGTCAGCTTCT
Mice β-actin	AGGATCACGACTGACAAAGGC	ATGGAGCCACCGATCCACA

## Data Availability

Data shall be made available on specific request.

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
