# Peer review of "Vindoline Exhibits Anti-Diabetic Potential in Insulin-Resistant 3T3-L1 Adipocytes and L6 Skeletal Myoblasts"

_nutrients, 2023, doi:10.3390/nu15132865_

Round 1

Reviewer 1 Report

Authors present the effects of Vindoline in an in vitro model of insulin resistance. There are some issues that require resolution before it is appropriate for publication.

Major comments:

The rationale for using dexamethasone then randomly switching to IL-1B is not sensible. IL-1B and dexamethasone would have very different effects in mediating insulin resistance. Authors should perform experiments with 1 compound or the other. Alternatively, they can utilize both for all measures, but not one of them for some measures and the other compound for other measures.

Overall the methods lack substantial detail to reproduce the data

Regarding figure 6. These images are very poor quality, with substantial differences in morphological features between groups (many apoptotic features). It is also highly recommended to have DAPI nuclear counterstain to confirm that staining is not nonspecific. Additionally, quantification should be provided. Fluorometric quantification in a black 96-well plate is preferable to images and is much more reliable.

Figure legends should contain sufficient information to stand alone from the paper without the need to search the text to understand methods. As such, legend details should be significantly expanded.

Regarding figure 5. Would you expect IRS-1 to decline, or modifications to postranslational sites? It is unclear the relevance of this mRNA data.

Minor comments:

Can authors provide the role of dexamethasone to give context to its dysfunction?

Which DMEM? provide a catalog number. What concentration of glucose? What about other additives that are often included in the DMEM formula (e.g. sodium pyruvate)?

Section 2.4 Minor error of 3t3 not t3

Line 84-85: Regardless of whether the protocol is established, these details should be provided.

Was mtt assessed with metformin and dexamthesone? This must be provided. Additionally, describe in detail the MTT procedures.

Line 116: the glucose oxidase method should be described in detail.

Line 148-149: the protocol should be described.

Data should be converted to standard deviation. Standard error is not appropriate in this context, as population means have no value, and it makes it difficult to assess the data appropriately. Update SD for all graphs as well.

Figure 1 is very blurry

Figure 2a is incomprehensible.

Figure arrangements are disorderly and need better structure.

Author Response

Comment-1: The rationale for using dexamethasone then randomly switching to IL-1B is not sensible. IL-1B and dexamethasone would have very different effects in mediating insulin resistance. Authors should perform experiments with 1 compound or the other. Alternatively, they can utilize both for all measures, but not one of them for some measures and the other compound for other measures.

Reply-1: The objective behind including these two models was to delineate the inflammatory aspect of a dysfunctional adipocyte induced by IL-1beta. However, as pointed out by the reviewer, inclusion of two model systems appears to create lack of clarity. Hence, we are removing the IL-1beta induced model results from the manuscript and are presenting only the dexamethasone-induced IR adipocyte data instead. The reviewer comment is accepted.

Comment-2: Overall the methods lack substantial detail to reproduce the data

Reply-2: We have revised and improved the methods section by including more details of the experimental conditions for better clarity and reproducibility. The reviewer comment is accepted.

Comment-3: Regarding figure 6. These images are very poor quality, with substantial differences in morphological features between groups (many apoptotic features). It is also highly recommended to have DAPI nuclear counterstain to confirm that staining is not nonspecific. Additionally, quantification should be provided. Fluorometric quantification in a black 96-well plate is preferable to images and is much more reliable.

Reply-3: As mentioned in reply-1, the data of IL-1beta induced inflammatory adipocytes have been removed from the manuscript. Hence, figure 6 and 7 are currently deleted in the resubmission.

Comment-4: Figure legends should contain sufficient information to stand alone from the paper without the need to search the text to understand methods. As such, legend details should be significantly expanded.

Reply-4: Figure legends have been revised and now incorporates more details to be supporting of figures as stand alone. The reviewer comment is accepted.

Comment-5: Regarding figure 5. Would you expect IRS-1 to decline, or modifications to postranslational sites? It is unclear the relevance of this mRNA data.

Reply-5: While several studies have reported on the protein expression of IRS1 and GLUT4 in diabetic condition, our study looked into the transcriptional suppression of these genes as a consequence of insulin resistance and the effect of vindoline in reversal of this condition. Previous studies have reported on the significant lowering of IRS1 mRNA in NIDDM individuals (CARVALHO, E., JANSSON, P.-A., AXELSEN, M., ERIKSSON, J.W., HUANG, X., GROOP, L., RONDINONE, C., SJÖSTRÖM, L. and SMITH, U. (1999), Low cellular IRS 1 gene and protein expression predict insulin resistance and NIDDM. The FASEB Journal, 13: 2173-2178. https://doi.org/10.1096/fasebj.13.15.2173) and in adipose tissue of abdominally obese women (Alain Veilleux, Karine Blouin, Caroline Rhéaume, Marleen Daris, André Marette, André Tchernof. Glucose transporter 4 and insulin receptor substrate–1 messenger RNA expression in omental and subcutaneous adipose tissue in women, Metabolism, Volume 58, Issue 5, 2009, Pages 624-631, ISSN 0026-0495, https://doi.org/10.1016/j.metabol.2008.12.007). Hence, we checked the mRNA expression of IRS-1 and GLUT4 genes to evaluate the antidiabetic efficacy of vindoline.

This explanation has been added to the discussion part of the revised manuscript.

Minor comments:

Comment-6: Can authors provide the role of dexamethasone to give context to its dysfunction?

Reply-6: Dexamethasone treatment prevent translocation of GLUT-4 to the cell surface via altering insulin signaling pathway by inhibiting phosphatidylinositol-3-kinase (PI3K) and serine/threonine protein kinase and induces insulin resistance. Dexamethasone treatment also enhances the activity of hormone sensitivity lipase, which leads to increased lipolysis in adipocytes (Sakoda H, Ogihara T, Anai M, Funaki M, Inukai K, Katagiri H, Fukushima Y, Onishi Y, Ono H, Fujishiro M, Kikuchi M, Oka Y, Asano T. Dexamethasone-induced insulin resistance in 3T3-L1 adipocytes is due to inhibition of glucose transport rather than insulin signal transduction. Diabetes. 2000 Oct;49(10):1700-8. doi: 10.2337/diabetes.49.10.1700. PMID: 11016454).

This information has been included in the manuscript. Reviewer comment is accepted.

Comment-7: Which DMEM? provide a catalog number. What concentration of glucose? What about other additives that are often included in the DMEM formula (e.g. sodium pyruvate)?

Reply-7: 3T3-L1 preadipocytes were cultured and differentiated to adipocytes in high glucose DMEM medium. L6 myoblast were cultured in high glucose DMEM medium. For differentiation from L6 myoblasts to myotubes, we have used low glucose DMEM medium.

Details of DMEM used is as follows- DMEM, High glucose: 4.5g/L D-glucose, L-glutamine, 110g/L sodium pyruvate (catalogue no. 11995065). DMEM, Low glucose: 1g/L D-glucose, L-glutamine, Sodium bicarbonate and 110g/L sodium pyruvate. Catalogue no. 12320032).

This information has been added to the manuscript. Reviewer comment accepted.

Comment-8: Section 2.4 Minor error of 3t3 not t3

Reply-8: We thank the reviewer for identifying the type error. This has been corrected in the revised manuscript.

Comment-9: Line 84-85: Regardless of whether the protocol is established, these details should be provided.

Reply-9: Rat skeletal muscle L6 myoblast cells were differentiated to L6 myotubes by the method of Gao et al., 2009. Briefly, myoblast cells maintained in high glucose DMEM medium supplemented with heat-inactivated 10% FBS plus antibiotic mixture (100U penicillin/ml and 100µg streptomycin/ml) were passaged at 80% confluence to differentiation medium. Differentiation was carried out in low glucose DMEM medium supplemented with 2% FBS for three days and the differentiation from L6 myoblasts to myotubes was confirmed by the formation of multinucleation in the cells.

We have added this section to the manuscript. Reviewer comment is accepted.

Comment -10: Was mtt assessed with metformin and dexamthesone? This must be provided. Additionally, describe in detail the MTT procedures.

Reply-10: MTT assay was not carried out for metformin and dexamethasone since previous studies have reported that 1µM dexamethasone and 1mM metformin is nontoxic (Marisol, M. M., Celeste, T. M., Laura, M. M., Fernando, E. G., José, P. C., Alejandro, Z., Omar, M. C., 439 Francisco, A. A., Julio César, A. P., Erika, C. N., Angélica, S. C., Gladis, F., Enrique, J. F., Gabriela, R. 440 Effect of Cucumis sativus on Dysfunctional 3T3-L1 Adipocytes. Sci Rep. 2019, 9: 13372)

MTT assay methodology: Cell viability of 3T3-L1 adipocytes and L6 myoblasts when treated with different concentrations (12.5, 25, 50, 75 and 100 µg/mL) was carried out in 96 well plates. Cells were incubated for 24 hours at under saturated humidity at 37⁰C and 5% CO2. After incubation period, media with aspirated, followed by addition of fresh media along with 20 µL MTT (5 mg/mL) solution/well. The yellowish MTT is reduced to dark coloured formazan by viable cells only. The formazan crystals formed were solubilized with MTT lysis buffer. The colour developed was quantified with an ELISA plate reader (BioRad systems, USA) (Measuring wave length: 570 nm). A graph was plotted by taking percentage viability in the Y-axis and concentration of vindoline in the X- axis.

This modification has been incorporated into the revised manuscript. Reviewer comment is accepted.

Comment-11: Line 116: the glucose oxidase method should be described in detail.

Reply- 11; Glucose oxidase kit method (Agappe Diagnostics, India). Briefly, 10µL sample or glucose standard was added to 1ml glucose reagent containing 92 mmol/ L tris buffer (pH 7.4), 0.3 mmol/L phenol, 15000 U/L glucose oxidase and 2.6 mmol/L 4-aminophenazone and incubated for 10 minutes at 370C. The absorbance was measured at 505 nm using microplate reader (Bio rad, USA).

Comment-12: Line 148-149: the protocol should be described.

Reply: The presence of high purity vindoline isolated from the leaf extract was confirmed by LC-MS/MS analysis (Shimadzu LC-MS/MS 8045) using a C18 column and compared with a digital library using Labsolutions software.

Comment-13: Data should be converted to standard deviation. Standard error is not appropriate in this context, as population means have no value, and it makes it difficult to assess the data appropriately. Update SD for all graphs as well.

Reply-13: Data has been converted to standard deviation format, as per reviewer’s suggestion.

Comment-14: Figure 1 is very blurry

Reply-14: We regret that the image is not very sharp. The analysis output obtained was of low resolution. We have made efforts to improve its clarity to some extent.

Comment-15: Figure 2a is incomprehensible.

Reply-15: Due to a technical error in uploading the manuscript, figure 2a got corrupt. We have replaced this with a fresh image in the revision.

Comment-16: Figure arrangements are disorderly and need better structure.

Reply-16: We regret the disorderly arrangement of figures. This has been rectified in the revision submission.

Reviewer 2 Report

The purpose of this study is to investigate vindoline on insulin resistance. However, it has been reported that vindoline (20 mg/kg) treatment significantly improved glucose homeostasis in db/db mice and STZ/HFD-induced type 2 diabetic rats, as reflected by its functions in increasing plasma insulin concentration, protecting the pancreatic β-cells from damage, decreasing fasting blood glucose and glycated hemoglobin (HbA1c), improving OGTT and reducing plasma triglyceride (TG). (J Ethnopharmacol 2013 Oct 28;150(1):285-97) In addition, Goboza, et al (2019) demonstrated that vindoline effectively ameliorated diabetes-induced hepatotoxicity by docking oxidative stress, inflammation and hypertriglyceridemia in type 2 diabetes-induced male wistar rats (Biomed Pharmacother2019 112:108638). On the other hand, a dose of 20 mg/kg body weight could potentially delay the progression of diabetes-related cardiovascular and kidney diseases via improving the antioxidant defence system and delay the initiation of apoptosis. Vindoline also exhibited excellent antihyperlipidaemic activities and could be utilised in the management of microvascular and macrovascular complications associated with diabetes (Biomedicines 2019 Aug 13;7(3):59). The effect of vindoline on insulin resistance (IR) models have been examined. Therefore, there is no novel finding in this study.

Author Response

Comment-1: The purpose of this study is to investigate vindoline on insulin resistance. However, it has been reported that vindoline (20 mg/kg) treatment significantly improved glucose homeostasis in db/db mice and STZ/HFD-induced type 2 diabetic rats, as reflected by its functions in increasing plasma insulin concentration, protecting the pancreatic β-cells from damage, decreasing fasting blood glucose and glycated hemoglobin (HbA1c), improving OGTT and reducing plasma triglyceride (TG). (J Ethnopharmacol . 2013 Oct 28;150(1):285-97) In addition, Goboza, et al (2019) demonstrated that vindoline effectively ameliorated diabetes-induced hepatotoxicity by docking oxidative stress, inflammation and hypertriglyceridemia in type 2 diabetes-induced male wistar rats (Biomed Pharmacother. 2019 112:108638). On the other hand, a dose of 20 mg/kg body weight could potentially delay the progression of diabetes-related cardiovascular and kidney diseases via improving the antioxidant defence system and delay the initiation of apoptosis. Vindoline also exhibited excellent antihyperlipidaemic activities and could be utilised in the management of microvascular and macrovascular complications associated with diabetes (Biomedicines 2019 Aug 13;7(3):59). The effect of vindoline on insulin resistance (IR) models have been examined. Therefore, there is no novel finding in this study.

Reply-1: As cited by the reviewer, previous works have investigated the antidiabetic properties of vindoline in STZ, HFD and other diabetic models. However, no systemic work has been carried out so far specifically addressing insulin resistance of adipocytes and skeletal myoblasts and the effect of vindoline on these specific pathophysiology. This study has also isolated vindoline in high purity from a natural source and evaluated in activity on two IR model systems which are of significance in addressing T2DM associated complications.

Round 2

Reviewer 1 Report

It is appreciated that authors have made edits which have improved the quality of the paper. I have a few additional comments to aid in the lack of clarity.

1. Indicate treatment times in figure legends.

2. Figure 2 states Mean and SEM, please modify to state SD

3. It is unclear what the other letters indicate with respect to statistical significance (c, d ,e) in the figures. I suggest this lettering scheme in the cited paper to make the results much more more clear (https://www.sciencedirect.com/science/article/pii/S0378874120336229#sec3)
Where values that do not share the same letter are significantly different from each other. This should be stated in statistical analysis section and the legends.

Author Response

We thank the reviewer for the meticulous review of the manuscript. All suggested corrections have been incorporated in the revised manuscript and highlighted for identification.

Reviewer 2 Report

There have been many complete animal study reports on the effect of vidoline on high-fat diet research in diabetes. The results were also similar to those of this study. Therefore, the research results of 3T3L1adipocyte in this research content should not be necessary. However, the mechanism of vidoline influence on sarcopenia is still unclear. Therefore, it is recommended that the authors can use skeletal myoblasts to investigate vidoline on sarcopenia more detail then resubmitting the manuscript.

Author Response

We thank the reviewer for the comments and suggestions made. 

Reply to comment:

  1. While the previous studies in animal models have investigated the effect of vindoline on streptozotocin induced wistar rats for T2DM, the model used was not specific for obesity associated adipocyte inflammatory diabetic complications. Also, the effect of vindoline on skeletal muscles are not well investigated. This study brings new understanding on the effect of vindoline in augmenting the glucose utilization capability and restoration of function of two key affected targets of T2DM, namely, adipose and skeletal muscles.  Therefore, this is not a repetition of already existing data. 
  2.  The idea of investigations on sarcopenia is very interesting. We shall extend our investigations on skeletal muscle cells for evaluating vindoline's effect on sarcopenia in our future investigations. 

However, as of now, we would like to present the existing data for addressing the effect of vindoline on adipocyte dysfunction and skeletal muscle pathophysiology in relation to obesity induced T2DM in our in vitro model studies, for publication. 

Round 3

Reviewer 2 Report

No more question.